# A Blockchain-Based Decentralized Public Key Infrastructure Using the Web of Trust

**Ratna Halder [†], Dipanjan Das Roy *,[†] and Dongwan Shin**

Department of Computer Science & Engineering, New Mexico Tech, Socorro, NM 87801, USA
* Correspondence: dipanjan.dasroy@alumni.nmt.edu
[†] These authors contributed equally to this work.

**Abstract:** Internet applications rely on Secure Socket Layer (SSL)/Transport Security Layer (TSL) certifications to establish secure communication. However, the centralized nature of certificate authorities (CAs) poses a risk, as malicious third parties could exploit the CA to issue fake certificates to malicious web servers, potentially compromising the privacy and integrity of user data. In this paper, we demonstrate how the utilization of decentralized certificate verification with blockchain technology can effectively address and mitigate such attacks. We present a decentralized public key infrastructure (PKI) based on a distributed trust model, e.g., Web of Trust (WoT) and blockchain technologies, to overcome vulnerabilities like single points of failure and to prevent tampering with existing certificates. In addition, our infrastructure establishes a trusted key-ring network that decouples the authentication process from CAs in order to enhance secure certificate issuance and accelerate the revocation process. Furthermore, as a proof of concept, we present the implementation of our proposed system in the Ethereum blockchain, confirming that the proposed framework meets the five identified requirements. Our experimental results demonstrate the effectiveness of our proposed system in practice, albeit with additional overhead compared to conventional PKIs.

**Keywords:** decentralized PKI; X.509; blockchain; Ethereum; smart contract; web of trust (WoT)

## 1. Introduction

In recent years, the rapid expansion of digital technology has led to an increased exchange of information, knowledge, and data over the Internet. As more and more devices, from wearables to motor vehicles, are now connected, and an increasing number of companies offer online services, there is a growing need for a secure environment to protect users' private information exchanged over the Internet. A PKI [1,2] is a widely adopted cryptographic approach that facilitates secure authentication, encryption, and integrity of communication. Furthermore, the SSL [3] is the standard security protocol that ensures secure connections for Internet traffic using a PKI and certificate chains. An SSL certificate is also pivotal in establishing an encrypted channel between the web server and the browser. Consequently, companies utilize SSL certificates on their web servers to establish secure connections, thereby fortifying the security of online transactions.

However, the entire SSL certificate issuance process is orchestrated through a centralized PKI, a hierarchical structure of CAs responsible for authenticating identities over the Internet [4]. CAs are entrusted with issuing public key certificates, ensuring cryptographic security by linking the public key to its owner. However, this over-reliance on one root authority often fails to maintain the transparency and security of the certificates as a whole.

As a result of the centralized PKI-based architecture, existing CAs have several drawbacks, including the following:

- The centralized structure of CAs creates a single point of failure, rendering them attractive targets for malicious third parties and posing potential risks to user data privacy and integrity [5,6].

- The existing system's distribution of revocation information is characterized by time-intensive processes and sluggish verification, which can result in prolonged revocation times during breaches [7].
- The lack of transparency in the current system enables CAs to issue certificates for individual domain owners without the parent company's knowledge [8], potentially leading to the creation of counterfeit sites.

Several mechanisms have been proposed to mitigate the issue of compromised CAs, as well as time-intensive revocation problems. These approaches have their own advantages and disadvantages compared to other schemes. One of the most popular approaches is the log-based schema [9,10], notably Certificate Transparency (CT) by Google, which detects fraudulent certificates and misbehaving CAs using a public log server to store and disseminate certified public key certificates, ensuring transparency and the ability to identify malicious activity. Alternative methods such as segmented CRL and Delta-CRL [11], as well as Pushing Revocation to Dependers [12], address challenges in certificate status distribution. Concurrently, various proposals have emerged advocating for a decentralized PKI, utilizing blockchain technology to enhance its infrastructure [13–19].

The centralized PKI used by existing CAs poses risks such as a single point of failure, lack of transparency, and vulnerability to adversaries, allowing a compromised CA to issue legitimate certificates for anonymous domains [8,20,21]. While a considerable amount of research is dedicated to swiftly detecting malicious attacks and compromised certificates, we adopted a different approach to address the underlying issues of a centralized PKI, particularly the problem of a single point of failure.

This study proposes a novel decentralized PKI that integrates a distributed verification system by leveraging blockchain and WoT [22] technologies. A high-level overview of the proposed system, incorporating blockchain and the WoT, is illustrated in Figure 1. The WoT technology facilitates the creation of a key-ring network comprising trusted entities responsible for authenticating certification information. The involvement of multiple members in the verification process eliminates single points of failure and Man-in-the-Middle (MitM) attacks [23], accelerates the revocation process, and the blockchain's append-only feature based on Markel Tree Hash enhances data transparency and public auditability. Furthermore, this paper delineates a comprehensive architecture for the distributed PKI, including the mathematical formulations necessary for computing the trust levels of both requesters and key-ring members, and also provides an implementation framework of the proposed system in the Ethereum blockchain [24]. The principal contributions of this paper are as follows:

- Introduction of a decentralized blockchain-based PKI system that ensures transparent issuance of X.509 certificates, thereby diminishing unauthorized activities.
- Integration of the WoT paradigm within the blockchain-based PKI to establish a trusted key-ring network, eliminating the need for central CAs.
- Empowerment of any ring member to serve as an auditor and initiate the revocation process upon detection of any malicious activities.
- Improvement of the algorithm and process for generating X.509 certificates to establish a robust digital certification system.
- Enhancement of the X.509v3 certificate extension field to include multiple signatures and blockchain-related information, facilitating seamless integration with existing browser certificate validation mechanisms with minor modifications.
- Facilitation of efficient computation of trust levels by incorporating the depth of the ring member nodes, which prioritizes proximity to the ring owner and maintains an optimal key-ring size.

The rest of this paper is organized as follows. Section 2 analyzes the evolution of blockchain-based PKIs. In Section 3, we discuss the open issues of the existing PKI mechanism, and Section 4 describes the technologies related to our work. We outline our proposal's system requirements and architectural model in Section 5. The implementation

details are presented in Section 6. Finally, in Sections 8 and 9, we conclude this paper with a discussion on possible future work on the blockchain-based PKI.

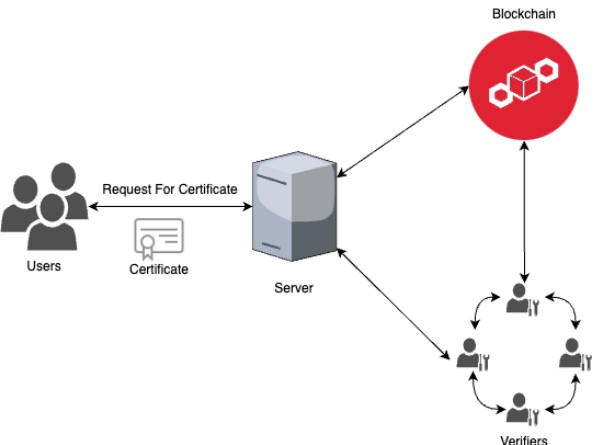

**Figure 1.** Overview of the proposed decentralized PKI based on blockchain and the WoT.

## 2. Related Works

Several previous studies have examined various certification services and digital identity systems using distinct approaches. The application of blockchain technology as a means to enhance PKI systems has emerged as a prevalent research focus within this domain.

Kubilay et al. introduced CertLedger [13], a blockchain-based PKI solution that leverages the blockchain as a public ledger for storing CA operations, certification, and revocation information. It helps mitigate split-world attacks, as clients can verify TLS certificates by monitoring logs post-confirmation of the latest block. However, CertLedger still struggles to prevent CAs from issuing rogue certificates. Building on the concept of transparency, the Decentralized PKI Transparency (DPKIT) system [14] was proposed to ensure the verifiability of certificates throughout their issuance and revocation life cycle. By facilitating public auditability, DPKIT enhances the ability to verify the authenticity of certificates and identify any fraudulent activities. Hwang et al. [15] presented a semi-decentralized PKI system that addresses the scalability challenge of managing a vast number of certificates. Their approach utilizes a TP-Merkle tree data structure, which, in combination with smart contracts and cryptographic evidence, enables an automated indemnification system. It also redefines the proof-of-violation (POV) protocol to achieve mutual nonrepudiation between the CA and the user and helps the domain monitor CA activities. BPKI [25] prevents domain name preemption attacks by introducing auditors and a multi-layered architecture to supervise CA operations and enhance blockchain transaction rates via delegated PBFT. While effective in preventing fraudulent certificate issuance and improving throughput, its reliance on multiple auditors can introduce delays in the certification process.

CBPKI [16] encompasses a cloud-based blockchain PKI that utilizes cloud computing and blockchain technology to enhance security, particularly in preventing DoS attacks by shifting CA operations to the cloud. It also improves performance by removing CRL. However, the certification phase remains under CA oversight, which perpetuates the threat of malicious certificate issuance when a CA is compromised. CertChain [17] is another decentralized and tamper-proof audit scheme built on Ethereum. It introduces a new data structure CertOper to store all certificate-related operation information. It also proposes a new entity bookkeeper, serving as a leader to manage certificate operations within the network. Users are required to validate certificates with the assistance of the bookkeeper, which introduces a security vulnerability: a compromised bookkeeper can exploit users to create a malicious certificate. In the quest for a fully decentralized authentication system, Certcoin [18] does not rely on trusted third parties for identity retention. It employs

the proof-of-work concept similar to Bitcoin, Namecoin, cryptographic accumulators, and distributed hash tables. In a similar vein, Qin et al. introduced Cecoin [19], which addresses the single point of failure associated with certificates on the Bitcoin network by replacing CAs with miners. Nevertheless, this method introduces unnecessary complexity to the revocation process when it is needed.

To detect misbehaving CAs or misused certificates, the log-based PKI approach was proposed. This approach uses a cryptographically secure, append-only log to maintain a record of legitimately issued certificates, thereby enhancing the revocation processes. Certificate Transparency (CT) [9] stands out as the most widely adopted log-based PKI system, utilizing append-only Merkle Hash Trees to manage and efficiently verify TLS certificates. CT allows for public monitoring of distributed logs, enabling oversight of CA operations and the management of compelled certificates by empowering legitimate domain owners to reject unauthorized certificates. This log-based approach was later adopted in many research works for building a transparent PKI, such as in enhanced certificate transparency [26], ARPKI [27], and CertLedger [13]. Yet, the log-based PKI approach still relies on current CAs and requires a continuous flow of data for storing records. Also, to revoke a certificate, action must be taken to communicate with the CA once any irregularities are detected in the CT log.

Another well-regarded solution that addresses the problems of a centralized PKI is a PKI that combines blockchain with the WoT. This WoT-based PKI ensures the authenticity of a public key without relying on a centralized authority, thus eliminating a single point of failure. DBPKI [28] is a dynamic blockchain-based PKI based on the WoT that distributes trust across various parties, thereby reducing dependence on traditional CAs and Certificate Revocation Lists (CRLs). However, as per the proposal, all the information would be recorded on the blockchain rather than providing digital signatures, indicating that the existing system for certificate verification, such as web browsers, would require substantial modifications in the authentication process.

BlockPGP [29] is an alternative blockchain-based PKI that creates PGP certificates and stores the public keys on a PGP key server. It integrates blockchain-related data into the PGP certificates, aligning with the existing OpenPGP standards. Also, existing certificate holders can sign other public key certificates. Similarly, Cecoin [19], SCPKI [30], LRS_PKI [31], and Trustful [32] are PKIs that combine WoT and blockchain technologies. Rather than relying on a centralized CA, these systems employ miners to fulfill the role of certificate authorities.

*Summary of Related Works*

According to our analysis, the majority of blockchain-based approaches incorporate blockchain into the existing PKI while adhering to the conventional CA structure. Certain proposals also suggest using miners as an alternative to centralized CAs, but they often lack comprehensive identity verification solutions. Table 1 outlines the various blockchain-based schemas, detailing their features, advantages, and limitations.

Additionally, a few other systems suggest significant modifications to the established certificate format. Our research distinguishes itself from existing works through its unique approach to certificate verification. We propose the construction of a trusted key ring comprising ring members who are responsible for the verification process. These members also monitor public records to identify and address any instances of misconduct or misuse of certificates. Furthermore, our system retains the standard X.509 certificate structure, with only minor additions to the extension fields, such as the inclusion of multiple signatures. This ensures compatibility with existing systems while enhancing the security and reliability of the certificate verification process.

**Table 1.** Comparison of the blockchain-based schemas.

| Schema | Features | Benefits | Drawbacks |
|---|---|---|---|
| Yakubov et al. [29] | A smart contract-based hybrid X.509v3 certificate that incorporates blockchain metadata. | • Mitigates compromised CA attacks.<br>• Updateable keys.<br>• Stores all data in the blockchain. | • Unable to detect counterfeit certificates.<br>• Does not hide identities. |
| CertLedger [13] | Certificate transparency and records all issued certificates and revocation status. | • Avoids split-world attacks.<br>• Ensures a reliable validation process. | • Reliance on CAs.<br>• Requires additional information to revoke a certificate. |
| Hwang et al. [15] | Leverages TP-Merkle tree data structure for transparency and a fully automated compensation system. Partially addresses scalability issues. | • Monitors the activities of CAs. | • Does not store all certificates.<br>• Extra storage necessary. |
| CBPKI [16] | Incorporates blockchain and cloud technologies. Decoupling data from the certificate authority | • Auto-scaling capability.<br>• Mitigates DoS attacks. | • Eliminates the necessity of CRLs, which raises usability concerns. |
| CertChain [17] | Establishes CertOper for storing certificates and enabling forward traceability. Segregates revocation history. | • Addresses centralization and block-traversal challenges.<br>• Utilizes an enhanced CertOper data structure. | • Increases complexity of revocation.<br>• A compromised bookkeeper could compromise security.<br>• Avoids using X.509 templates. |
| CertCoin [18] | Manages a public record of domains and associated public keys utilizing Bitcoin, distributed hash tables, and a Merkle accumulator as a CRL. | • Updateable public keys.<br>• Fully public and auditable. | • Employs larger block sizes.<br>• Exhibits logarithmic complexity for new CRLs. |
| CeCoin [19] | Blockchain and web-of-trust-based PKI influenced by Bitcoin. Miners replace CAs. | • Prevents a single point of failure.<br>• Eliminates central CAs. | • Lack of information about the verification process by miners. |
| DBPKI [28] | A dynamic PKI based on blockchain and the WoT. Any system entity can serve as an auditor and handle revocation tasks. | • Eliminates a single point of failure and removes the need for CAs. | • Additional cost overhead for storing revocation data in blockchain. |
| SCPKI [30] | Utilizes Ethereum smart contracts to detect malicious certificates and the WoT for verification. Saves certificate data in the InterPlanetary File System (IPFS). | • IPFS reduces gas costs. | • Introduces additional latency on certificate authentication. |
| META_PKI [33] | A cross-certification approach employing a smart contract on Hyperledger Fabric. It offers distributed authentication by using multiple service providers. | • Eliminates trust in a central CA.<br>• Advanced log-based system. | • Unable to prevent malicious certificate issuance.<br>• Lacks complete transparency. |
| Proposed system | WoT model and blockchain-based PKI. Incorporates trusted key rings and an advanced verification system. Offers faster revocation. | • Eliminates a single point of failure, MitM attacks, and no central CA.<br>• Fully transparent and monitors CA activity.<br>• Prevents malicious certification. | • Auto-scaling not supported for larger networks. |

## 3. Shortcomings of a Centralized PKI System

I. **Single Point of Failure:** The centralized nature of CAs presents a significant vulnerability, leading to widespread distrust, regardless of their theoretical trustworthiness. Multiple operational errors and breaches in well-known CAs have raised significant concerns [8,34], such as the 2017 Symantec incident, where over a hundred certificates were improperly issued, resulting in distrust from major platforms [5].

The large number of CAs and their dependent hierarchy also pose a significant challenge, as a single compromised or rogue CA among the hundreds or thousands could lead to widespread failure, often exploited through MitM attacks [20,34]. In addition, compromised CAs can be exploited for malicious purposes, potentially undermining the entire infrastructure and compromising sensitive data; they are also able to issue a legitimate certificate for any domain [6,21,34].

II. **Challenges in Certificate Revocation Process:** Revoking specific certificates issued by a CA is essential for various reasons, such as compromised private keys, compromised CAs, change of affiliation, or certificate supersession [7,35]. The MITRE Corporation highlights the expense and time-consuming nature of a centralized PKI's revocation information distribution, particularly in revoking certificates before expiration [1].

One common method for revoking certificates involves Certificate Revocation Lists (CRLs), managed by the CA or other trusted parties, where the issuing CA publishes revoked certificates with their serial numbers [2]. However, if the CRL becomes unreachable, clients may experience either an open or closed failure. Additionally, CAs, even in the event of a breach, may not be entirely trustworthy in containing or reporting incidents due to financial incentives and reputational concerns, often choosing not to disclose data breaches or operational errors.

III. **Lack of Transparency:** In the current system, a CA can issue a certificate for any individual domain owner. However, the parent company does not have any way of knowing whether any individual domain owner is receiving a certificate with the parent company's domain name, potentially enabling the issuance of fake certificates for deceptive purposes [8].

Currently, the only viable option is to maintain a log of all issued certificates using a log-based schema, accessible for client verification. Another potential method involves a database containing all possible CAs capable of issuing certificates for specific domains, although this approach is currently not widely adopted.

In light of these limitations, it is evident that the existing centralized PKI architecture presents significant vulnerabilities and inefficiencies. Addressing these challenges is paramount to ensure the continued security and integrity of online communications and transactions.

## 4. Background Technologies

### 4.1. Blockchain and Ethereum

Blockchain is a distributed ledger technology that allows for the secure, transparent, and tamper-proof recording of transactions. Validators, or miners, confirm transactions by solving complex mathematical problems, with the specific validation method depending on the blockchain (e.g., proof-of-work or proof-of-stake). Each transaction is timestamped and cryptographically linked to the previous block, forming an unalterable chain of records. This immutable characteristic ensures that blockchain records are distributed, shared, and maintained across a peer-to-peer network, making them highly secure and reliable.

Ethereum is a well-established, open-source decentralized blockchain platform that enables the creation and deployment of smart contracts and decentralized applications (DApps). Developers have the flexibility to establish their own rules for executing transactions, including custom transaction formats and state transition functions. The platform incorporates key blockchain principles such as a decentralized database, transparency, security, and efficiency, ensuring that all stored information is safeguarded against deletion and tampering by malicious actors.

Ethereum smart contracts are autonomous programs that execute transactions when certain conditions are fulfilled. While these contracts can be coded in low-level, stack-based bytecode referred to as EVM code, the primary method involves using high-level languages like Solidity. Solidity-written smart contracts are compiled to EVM code for execution on the Ethereum Virtual Machine. Ethereum also supports popular languages such as C, Java, Python, etc.

Smart contracts come with several advantages compared to traditional computer programs, including autonomy, as their execution is overseen by the network; trust, validated through consensus among nodes; data security, as the application's data are permanently stored in the blockchain; and transparency, as the code and storage of smart contracts are publicly available. Smart contracts also help manage access rights and enforce agreements, ensuring that other contracts cannot make unauthorized changes. Aligning seamlessly with our proposed system, smart contracts are crucial for following a set of predefined business rules that trigger different actions based on specific conditions. For example, if trust levels reach certain thresholds or when specific requirements are met, smart contracts enable the execution of corresponding actions. Smart contracts also provide a read-only mode for

accessing data and checking smart contract status without altering the blockchain. This feature is cost-free and aids in constructing a transparent system accessible to the public.

### 4.2. Web of Trust

The Web of Trust (WoT) is a decentralized alternative to the traditional role of CAs in the PKI. In the WoT model, participants are empowered to declare their trust in fellow members, thereby enabling them to introduce and authenticate a new public key by signing with their private key. Consequently, each network member can act as a potential CA. The level of trust can be divided into four categories: full, marginal, no trust, and unknown. This method of establishing trust parallels the *chain of trust* seen in conventional systems.

Pretty Good Privacy (PGP) is a cryptographic protocol that adopts the WoT's distributed trust framework. Network participants validate certificates based on trust levels and sign one another's keys. This process gradually forges a Web of Trust, where individual public keys are interlinked by these signatures [36]. PGP employs a formulaic approach to represent the validation process, denoted as key legitimacy $L = c/C + m/M$, where $c$ and $m$ denote the count of signatures from fully and marginally trusted users, respectively [22]. $C$ represents COMPLETES_NEEDED, and $M$ symbolizes MARGINAL_NEEDED. We integrated this PGP validation method to ensure certificate authenticity within our key-ring network.

### 4.3. X.509 Certificate

X.509 is the standard for certificates used in SSL/TLS protocols [2] to establish secure connections by binding entities—such as users, organizations, or devices—to a public key, thereby ensuring privacy and authenticity. An X.509 certificate includes the subject's details, issuer information, and key details such as the version number, serial number, signature algorithm, validity period, and the subject's public key [4,37]. Version 3 of X.509 introduced an extension field that allows for the inclusion of additional standardized information. Although there are standard extensions, the system is flexible, allowing for new extensions to be registered with regulatory bodies like the ISO. These extensions are categorized into groups related to key information, policy information, user and CA attributes, and certification path constraints, enhancing the certificate's functionality and security.

The certification path constraint extensions in X.509v3 certificates allow a CA to manage and restrict the level of trust extended to third parties in a cross-certified setup. These extensions are divided into three fields: basic constraints, name constraints, and policy constraints. Each X.509v3 certificate contains three attributes to define these extensions:

- **extnID:** Denoting Extension ID, which is an Object Identifier (OID) defining the extension's definition and format.
- **critical:** A boolean indicating the importance of the extension.
- **extnValue:** The actual value of the extension.

Below is the ASN format for the extension field [2,4]:

```
Extensions  ::=  SEQUENCE SIZE (1..MAX) OF Extension
Extension   ::=  SEQUENCE {
    extnID       OBJECT IDENTIFIER,
    critical     BOOLEAN DEFAULT FALSE,
    extnValue    OCTET STRING
                 - contains the DER encoding of an ASN.1 value
                 - corresponding to the extension type identified
                 - by extnID
}
```

The *critical* flag signals whether an extension is essential; if set to true, the extension must be recognized by the verifying application for the certificate to be valid, while non-critical extensions can be disregarded. Our presented architecture incorporates the use of an extension field to better support the inclusion of multiple signatures from ring

participants. It is crucial that this proposed extension is marked as mandatory, ensuring that web browsers recognize our custom extension field. While the incorporation of a unique extension field is possible, its implementation should provide significant benefits to be deemed important.

## 5. System Overview

### 5.1. System Requirements

In order to construct a decentralized SSL certificate system, a set of fundamental requirements was identified. These requirements serve as the guiding principles for the proposed system and are outlined as follows:

1.  **User Validation:** Ensures that only legitimate domain owners request SSL certificates by authenticating their identity to create an X.509 certificate. It is crucial for introducers to verify and confirm that only authorized domain owners are granted digital certificates.
2.  **Decentralization:** Involves multiple ring members, also referred to as introducers, to avoid a single point of failure. Allows authorized introducers to authenticate certificate information and add individual signatures, resolving dependency on the existing CA hierarchy.
3.  **Transparency:** Publicly accessible certificate creation and verification data are stored on the blockchain network. This facilitates public visibility of certificate issuances and revocations, enabling the detection of certificates that have been tampered with or compromised introducers.
4.  **Interoperability:** Requesters are provided with X.509v3 format certificates to ensure privacy, legitimacy, and secure connections during TLS handshakes. Ensuring compatibility with the existing browser verification mechanism is essential, necessitating minor adjustments.
5.  **Revocation Efficiency:** The certificate revocation process should be expedited. The system will automatically recognize and respond to revocation requests from authentic users or introducers. Additionally, it is pivotal to engage multiple introducers to thoroughly authenticate the revocation request.

### 5.2. System Architecture

In light of the fundamental requirements delineated in the preceding section, the proposed system design encompasses four pivotal building blocks: (I) certificate issuance, which entails receiving user requests and initiating a blockchain transaction; (II) X.509 certificate creation, involving the construction of X.509v3 certificates and enabling introducers to append signatures; (III) information validation, focusing on authenticating all user-provided information; and (IV) key-ring establishment, dedicated to establishing a key-ring network based on trust. The subsequent section offers an in-depth description of each module.

#### 5.2.1. Certificate Issuance

In order to request a certificate, the client is required to generate a Certificate Signing Request (CSR) and digitally sign the information using the client's private key. Subsequently, the entire package, along with the corresponding public key, is transmitted to the blockchain network for initial verification. The CSR contains essential details such as the domain name, country code, state/locality, organization name, department name, email address, and optional trusted users. Notably, the domain name serves as the subject name, while the email address must be an official organization email for domain verification and future communication with the requester. Following this, the CSR information and the associated public key undergo initial verification within the blockchain network. Figure 2 depicts a high-level overview of creating a signed CSR using the requester's public key and integrating the request into the network for the initial verification process.

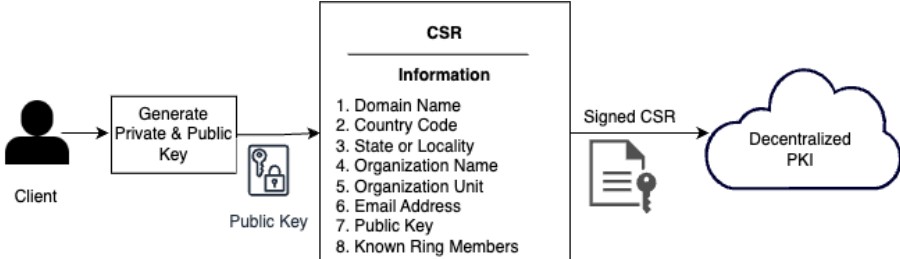

**Figure 2.** Illustration of the process of generating a signed CSR and integrating it into the blockchain network.

This validation process includes confirming the digital signature using the provided public key to ensure the integrity of the information, thereby verifying that the CSR information remains unaltered. Furthermore, an automatic validation process is conducted on the domain name and other CSR details. Upon successful validation, an X.509 certificate is generated and shared with the key-ring network for subsequent validation.

5.2.2. Validation

Ensuring the authenticity of CSRs stands as a pivotal aspect of the certification process. To achieve this, our proposed decentralized PKI system introduces multiple introducers within the verification system. The initial phase involves identifying validators responsible for information validation, followed by the establishment of a key-ring network comprising trusted introducers, who will later undertake information verification. In summary, the system involves two types of validation: (I) public key certificate authenticity, focusing on information validation for issuing/revocating certificates, and (II) ring-member authenticity, which is applicable to introducer validation for their inclusion within the key-ring network.

(A) **Public Key Certificate Authenticity:**

In a decentralized PKI structure, there is no central authority for certification, so validated introducers act as CAs and sign certificates to ensure trustworthiness. Figure 3 demonstrates the process by which certificates can be authenticated by trusted introducers, bypassing the need for CAs. These members manually verify all information, requesting additional details or physical meetings when necessary, and document their verification methods within the signed certificates, thereby enhancing trust in user information for both the introducers and other ring members. During the process of signing a certificate, introducers are required to assess and define the level of trust for the examined certificate after verifying the CSR information. The PGP trust model relies on varying levels of trust determined by the introducers during certificate verification. We employ a similar approach to calculate the trust level of the certificate. The trust level assigned to a certificate falls into one of the following categories:

- Not trusted;
- Marginally trusted;
- Fully trusted.

Once the manual validation and signing process is completed by introducers, the system examines the number of signatures and their associated trust levels. It then employs a mathematical formula that takes into account several parameters to calculate the overall trust level. The mathematical equation used to determine the trust level of the public key certificate is as follows:

$$L = \frac{1}{C} \sum_{i=1}^{c} \frac{1}{d_i} + \frac{1}{M} \sum_{i=1}^{m} \frac{1}{d_i} \tag{1}$$

In Equation (1), the following variables are defined:

- *L* represents the certificate trust level.
- *c* and *m* represent the number of signatures from fully trusted and marginally trusted introducers, respectively.
- *C* and *M* denote the number of signatures required from fully trusted and marginally trusted introducers to validate the certificate. These values are determined by the system authority, where $C \geq 1$ and $M \geq 1$.
- *d* represents the depth from the ring owner to the current introducer. In graph terms, the depth signifies the distance from the root node to the current node. Here, calculating the depth value allows us to define the closest to the furthest introducers in the ring hierarchy based on their distance from the ring owner. A depth constraint can also be imposed to reduce the expansion of the key-ring size.

Based on the values obtained from introducers and utilizing the aforementioned mathematical equation, the public key certificate is deemed fully valid if $L \geq 1$, marginally valid if $0 < L < 1$, and unknown if $L = 0$. For instance, by setting $C = 1$ and $M = 3$, indicating the requirement of at least one signature from a fully trusted introducer or at least three signatures from marginally trusted introducers, and using a depth *d* of 1 for all the signers, the certificate will be fully trusted; otherwise, it will be marginally or not trusted. This trust level is subsequently appended to the requester's certificate, offering transparency for individuals from other communities to comprehend the level of trust they should place in it.

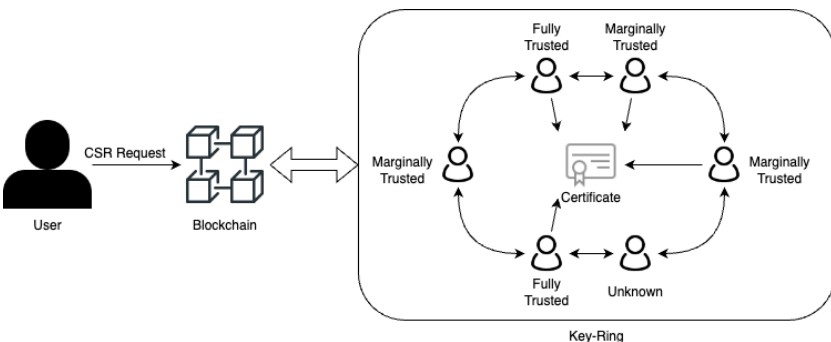

**Figure 3.** A diagram showcasing the participation of key-ring members with different trust levels in verifying a certificate.

(B) **Ring-Member Authenticity:**
The validation of a certificate relies on the introducer's signature, emphasizing the importance of introducers being trusted with valid keys when establishing a key ring. In the system, existing members can verify and sign new members' public keys to create bridges within the key-ring community. Analogous to certificate verification, a trust level is assigned during the validation process of new members. The trust level of an introducer can be categorized as *fully trusted*, *marginally trusted*, or *not trusted*. These trust levels are factored into the final calculation. For instance, a fully trusted introducer may assign a marginally trusted level to a new member if necessary, while a marginally trusted introducer can assign a fully trusted level during signing. Additionally, each introducer has a depth level along with their level of trust. Figure 4 shows an example of calculating the depth levels of introducers, whether they are directly or indirectly connected to the ring owner. For instance, introducers directly connected have a depth of one, while those indirectly connected have a depth greater than one, varying based on distance. Similar to the verification of public key certificates, a mathematical equation is employed to compute the trust level for the newly introduced.

The equation for calculating the trust level during introducer validation is as follows:

$$T = \frac{c}{C} + \frac{m}{M} + \frac{n}{N} \tag{2}$$

$$c = \sum_{i=1}^{x} \frac{c_c}{d_i} + \sum_{i=1}^{y} \frac{c_m}{d_i} + \sum_{i=1}^{z} \frac{c_n}{d_i} \tag{3}$$

$$m = \sum_{i=1}^{x} \frac{m_c}{d_i} + \sum_{i=1}^{y} \frac{m_m}{d_i} + \sum_{i=1}^{z} \frac{m_n}{d_i} \tag{4}$$

$$n = \sum_{i=1}^{x} \frac{n_c}{d_i} + \sum_{i=1}^{y} \frac{n_m}{d_i} + \sum_{i=1}^{z} \frac{n_n}{d_i} \tag{5}$$

In Equation (2), the following variables are defined:

- $T$ represents the introducer's trust level.
- $c_c$, $c_m$, and $c_n$ denote the full, marginal, and not trusted trust levels given by fully trusted introducers. $c_c$, $c_m$, and $c_n$ are the constant weights determined by the system authority, where, $c_c > c_m > c_n$. Similarly, $m_c$, $m_m$, and $m_n$ are the constant weights for the full, marginal, and not trusted trust levels given by marginally trusted introducers. The same goes for $n_c$, $n_m$, and $n_n$ for not trusted introducers.
- $c$ denotes the sum of all trust level weights from the fully trusted introducers divided by their respective depth $d$. Similarly, $m$ represents the sum of all trust level weights from the marginally trusted introducers divided by their respective depth $d$. The same goes for $n$ not trusted introducers.
- $C$, $M$, and $N$ denote the number of signatures required from the fully, marginally, and not trusted introducers to validate new introducers. These values are determined by the system authority where $C \geq 1$, $M \geq 1$, and $N \geq 1$.

The introducer is classified as fully trusted if $T \geq 1$, marginally trusted if $0.5 \leq T < 1$, and not trusted if $T < 0.5$. Upon successful validation, the new introducer will be incorporated into the key-ring network, and an announcement will be broadcast over the network to inform existing key-ring members.

This section outlines various scenarios to explain the trust evaluation mechanism. For illustration purposes, in all scenarios, the weights assigned to the constants are as follows: $C = 2$, $M = 3$, and $N = 4$; $c_c = 4$, $c_m = 3$, and $c_n = 2$; $m_c = 3$, $m_m = 2$, and $m_n = 1$; and $n_c = 2$, $n_m = 1$, and $n_n = 0$.

**Scenario 1:**

- Two endorsements are received from fully trusted introducers (one fully trusted, one marginally trusted).
- Three endorsements are received from marginally trusted introducers (two fully trusted, one marginally trusted).
- The trust score is $T = 4.4722$ and $T > 1$. Hence, the new participant is deemed a fully trusted introducer.

$$c = \frac{4}{1} + \frac{3}{2} \tag{6}$$

$$m = \frac{3}{1} + \frac{3}{2} + \frac{2}{3} \tag{7}$$

$$n = 0 \tag{8}$$

$$T = \frac{c}{C} + \frac{m}{M} + \frac{n}{N} = 4.4722 \tag{9}$$

**Scenario 2:**

- One endorsement is received from a fully trusted source (not trusted).
- Two endorsements are received from marginally trusted sources (marginally trusted).

- Two endorsements are received from untrusted sources (marginally trusted).
- $T = 0.9250$ and $0.5 <= T < 1$, marking the participant as marginally trusted.

$$c = \frac{2}{4} \tag{10}$$

$$m = \frac{2}{4} + \frac{2}{5} \tag{11}$$

$$n = \frac{1}{1} + \frac{1}{2} \tag{12}$$

$$T = \frac{c}{C} + \frac{m}{M} + \frac{n}{N} = 0.9250 \tag{13}$$

**Scenario 3:**

- One endorsement is received from a marginally trusted source (not trusted).
- Three endorsements are received from untrusted sources (two marginally trusted, one not trusted).
- $T = 0.4417$ and $T < 0.5$. Therefore, the new participant is considered an untrusted introducer.

$$c = 0 \tag{14}$$

$$m = \frac{1}{5} \tag{15}$$

$$n = \frac{1}{1} + \frac{1}{2} + \frac{0}{3} \tag{16}$$

$$T = \frac{c}{C} + \frac{m}{M} + \frac{n}{N} = 0.4417 \tag{17}$$

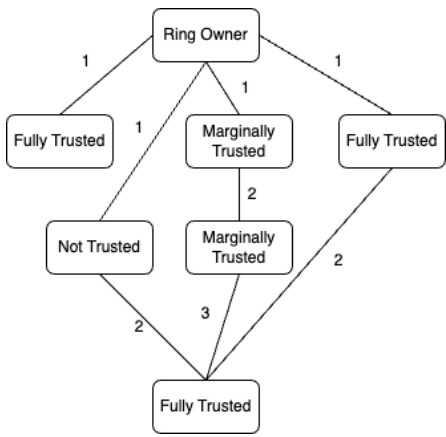

**Figure 4.** A representation of a key-ring network with each member's depth and trust level.

5.2.3. X.509 Certificate Creation

An X.509v3 certificate introduces an extension field, providing a standardized and generic method for including additional information. This flexibility in the extension field enables the addition of multiple signatures, smart contract information, and trust levels, among other features, as illustrated in Figure 5.

In contrast to the conventional CA hierarchy, the proposed certificate format diverges from the traditional path, eliminating the need to maintain a certificate path from the issuing or intermediate CA to the root CA [1]. Instead of a single signature from a certificate authority, the proposed format mandates multiple signatures, each obtained from the introducer, who verifies the certificate content. This approach significantly reduces the probability of issuing certificates to illegitimate users, as compromising multiple introducers presents a substantially greater challenge compared to compromising a single

CA. As previously mentioned, an X.509v3 certificate contains some extra information on extension fields, as follows:

- **Multiple Signatures:** Accommodates signatures from multiple introducers.
- **Certificate Trust Level:** Contains the trust level calculated using Equation (1) during certificate validation.
- **Transaction ID:** This is a unique ID for blockchain transactions and is used to fetch certificate-related information from the blockchain network.
- **Introducer Identifier:** This identifier is used as a unique address to identify introducers in key-ring smart contracts.

| |
|---|
| X.509 version number |
| User public key |
| Serial number |
| Subject name |
| User unique identity |
| Validity period |
| *Ring owner* unique identity |
| **Extension** |
| Introducer-1 signature |
| Introducer-2 signature |
| … |
| *Certificate trust level* |
| *Transaction ID* |
| *Introducer Identifier* |

**Figure 5.** X.509v3 format with multiple signatures and blockchain information in the extension field.

Abstract Syntax Notation One (ASN.1) [11] serves as a standard interface description language (IDL) for defining data structures and is employed to accommodate the proposed custom extension field. The ASN.1 structure encompasses an extension identifier, a signature algorithm, a signature value, and a critical flag. The critical flag denotes the significance of the extension information. If the flag value is true, the verifying application must recognize the extension; otherwise, the certificate cannot be accepted. However, an application can ignore the extension field if it is not marked as critical. For the proposed system, the extension field is utilized to support multiple signatures signed by introducers. It is imperative that this extension be marked as critical to ensure recognition by web browsers. Below is the pseudo-ASN.1 structure with notations and the textual description outlining the proposed extension field for X.509v3.

```
bCertInfo EXTENSION ::= {
  SYNTAX            BCertInfoSyntax
  IDENTIFIED BY     id-ce-bCert
}

BCertInfoSyntax ::= SEQUENCE SIZE (1..MAX) OF~BCertInfoSequence

BCertInfoSequence ::= SEQUENCE {
  cA                BOOLEAN DEFAULT TRUE,
  signatureAlgorithm    AlgorithmIdentifier{{SupportedAlgorithms}},
  signatureValue        BIT STRING
}

multiSignExtension  OBJECT IDENTIFIER   ::= {joint-iso-itu-t asn1(1)
                                            ber-derived(2)
```

```
                                          distinguished-encoding(1)}
id-ce              OBJECT IDENTIFIER   ::= multiSignExtension
id-ce-bCert        OBJECT IDENTIFIER   ::= {id-ce bCertKey}
bCertKey                               ::= INTEGER(0..MAX)
```

Each extension is linked with an Object Identifier (OID), as defined in X.509, which serves to identify the extension. This OID is part of the *id-ce arc*. The *signatureAlgorithm* field indicates the identifier for the cryptographic algorithm employed by the introducers to sign the X.509 certificate. Distinguished Encoding Rules (DER) encoding is utilized here to define the signature context. The *signatureValue* field contains the digital signature encoded in the ASN.1 DER as BIT STRING. Details of this field are provided in RFC3279 [38], RFC4055 [39], and RFC4491 [40].

For multi-signature extension *multiSignExtension*, the OID of the ASN.1 DER encoding is defined in the ITU-T X.680 standard [41]. In this standard, the text preceding each integer describes its contextual meaning, while the integer itself represents the encoding of that particular meaning.

By employing this ASN.1 structure, additional information, such as multi-signatures, can now be accommodated within the X.509v3 extension field. Similarly, this field can also incorporate other information, including the certificate trust level, transaction ID, and introducer identifier. The presence of this extension field can be identified by its respective OID.

### 5.2.4. Certificate Revocation

Certificates require revocation upon expiration, yet certain circumstances may demand invalidation before their expiry, such as private key compromise, compromised introducer-provided certificates, or domain-name changes. To ensure timely and accurate revocation, our system meticulously examines all revocation requests, engaging multiple ring members in the verification process. With sufficient evidence, the certificate is revoked by the ring members. Authorized members, including certificate holders, existing ring members, regular users, and government entities, may request certificate revocation for various reasons.

Revocation requests may stem from the following sources:

- **Certificate Owners:** They may request revocation in the event of server attacks or upon suspicion of compromised, stolen, or lost private keys.
- **Authenticated Ring Members:** These members possess the authority to request revocation for diverse reasons.
- **Other Entities:** This category encompasses users who have evidence of certificate misuse, deception, or inappropriate behavior, as well as authorized government members who can apply for certificate revocation.

When submitting a revocation request, the requester must provide a valid reason and supporting identity authentication documents, such as a signed X.509 digital certificate and public key. This signature ensures the authenticity of the request, while the public key serves as evidence of possession of the private key and other pertinent information.

When a revocation request is received within the blockchain network, introducers are given priority to initiate the request verification process. The system automatically conducts signature validation to confirm the requester's authenticity using the provided public key. Introducers are required to validate additional information, such as the reason for revocation, which may necessitate manual validation and subsequent signing. The system employs a process akin to the X.509 certificate validation process and assesses a mathematical equation (Equation (1)) to determine whether to invalidate the certificate. Subsequently, the ring owner confirms certificate revocation by responding to the requester and also updates the Online Certificate Status Protocol (OCSP) [42] server to reflect the changes in the revocation list. Browsers can verify the revocation status of a certificate by utilizing the Certificate Revocation List (CRL) or processing the OCSP response.

Additionally, the proposed system stores revocation information in the blockchain network, thereby ensuring public accessibility to this information. This approach allows anyone to promptly access the revocation details. The key-ring network, as proposed, is established with multiple trusted introducers, thereby enhancing the likelihood of having available introducers for prompt validation of revocation requests. The involvement of multiple introducers concurrently enhances the system's efficiency and security.

*5.3. Blockchain Integration*

Involving multiple introducers in the verification process reduces the risk of a single point of failure, although it does not entirely prevent tampering with existing certificate information. Making certificates and introducer information publicly visible, without the concern of tampering, facilitates the quick detection of malicious members and certificates. This also expedites the certificate revocation process and eliminates malicious introducers from the key ring. Implementing the proposed system in a blockchain environment ensures a distributed nature and public visibility of system information. Blockchain technology offers an immutable chain of records and maintains it across multiple nodes in a peer-to-peer (P2P) network, thereby ensuring the integrity of existing records in the system.

Ethereum, an open-source blockchain platform, encompasses most of the key principles of the blockchain infrastructure and offers a mature development ecosystem. It enables the creation of decentralized applications across various domains. The preferred approach is to develop the proposed system using Ethereum smart contracts to achieve decentralization and ensure public visibility. This approach not only aligns with the goals of the proposed system but also leverages the robust infrastructure and capabilities of Ethereum to establish a secure and transparent decentralized PKI.

## 6. Implementation

Our implementation aims to test the feasibility of our proposed system by maintaining a frontend and backend architecture, with the backend operating on a decentralized peer-to-peer Ethereum blockchain. The high-level decentralized application architecture, along with the associated tools and technologies, is illustrated in Figure 6. The frontend is responsible for enabling user account creation and managing CSR requests, whereas the backend is tasked with maintaining Ethereum smart contracts and storing data within the blockchain network.

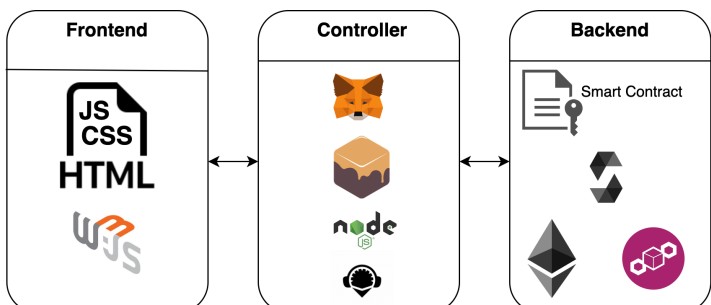

**Figure 6.** Resources for system development and testing.

Smart contracts form the core of the Ethereum blockchain, and our backend comprises three smart contracts:

- **Issue_cert:** Responsible for processing CSR and revocation requests, as well as initial verification.
- **Certificate:** Generates X.509 certificates and oversees the signing process.
- **Verification:** Manages verification processes with introducers and aids in building the key ring.

Additionally, the backend system incorporates a controller built with Node.js, an Ethereum client, and an SMTP server. It enables communication between the client side and

smart contracts, ensuring seamless interaction throughout the system. Figure 7 provides a detailed illustration of the step-by-step implementation workflow and the role of each smart contract.

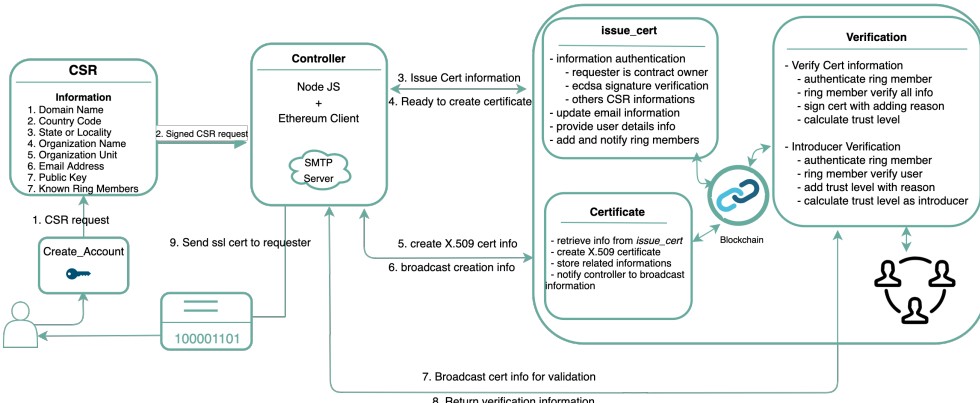

**Figure 7.** Implementation workflow of the proposed architecture.

### 6.1. Certificate Signing Request

The Ethereum platform creates a supportive environment for development in JavaScript because we utilize jQuery and Node.js to create web applications. This platform is also equipped with the Web3 JavaScript API, which empowers developers to construct client-side applications capable of engaging with the blockchain via smart contracts. For the issuance of certificates, it is imperative for users to establish an Ethereum account, which encompasses a private key, a public key, and an Ethereum address.

Our proposed system streamlines the generation of private keys and Ethereum addresses by leveraging the capabilities of the Web3 API. Furthermore, the retrieval of the public key from the private key is executed through the EthereumJS API. It is important to acknowledge that within the Ethereum ecosystem, the public and private key pair remains constant across different networks.

Upon the creation of a valid Ethereum account, users are enabled to submit requests for certificate issuance through a CSR form, as presented in Figure 8a. This form mandates the inclusion of specific fields, such as domain name, country code, state, organization name, organization unit, and email address, while also accommodating optional information.

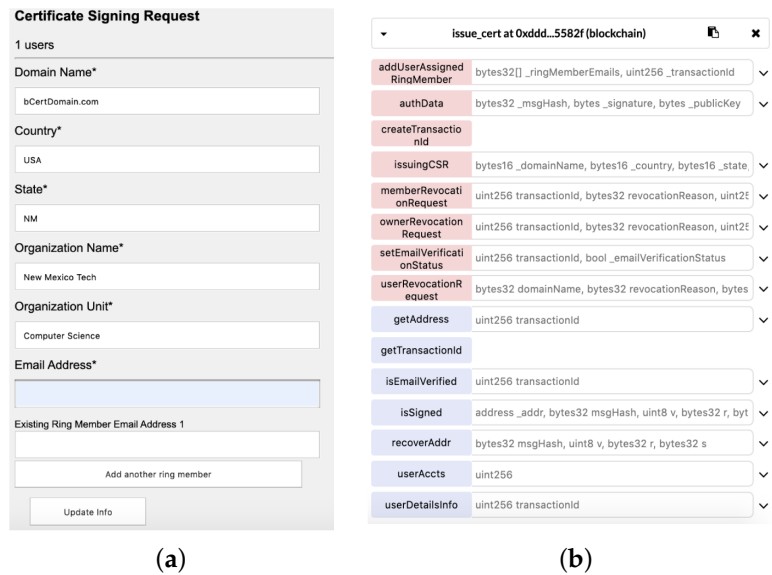

(**a**)          (**b**)

**Figure 8.** Certificate signing request using a frontend UI (**a**) and also using *issue_cert* functions from Remix IDE (**b**). Note: * is a mandatory field in CSR form

## 6.2. Backend Controller

The controller serves as a part of the system's backend architecture, with its primary responsibility being the facilitation of communication between the requester and the system. This communication encompasses a variety of functions, including the dispatch of verification emails, the issuance of certificates, and the synchronization of information with smart contracts on the Ethereum blockchain. Upon receipt of a CSR, the controller initiates a verification process by sending an email to the address specified within the CSR. Subsequent to email verification, the controller proceeds to secure the CSR information by employing the SHA-3 cryptographic hash function to generate checksums. The hashed CSR data are then digitally signed utilizing the requestor's private key in accordance with the Elliptic Curve Digital Signature Algorithm (ECDSA). Following the generation of the digital signature, the controller is tasked with the synchronization of the signed and hashed CSR with a designated smart contract on the Ethereum blockchain, named *Issue_cert*. The interaction with the smart contract is mediated through web3.js.

Figure 7 provides a visual representation of the controller's role within the broader system infrastructure, delineating its interconnections with other system components and illustrating the flow of information from the CSR submission to the blockchain.

## 6.3. Smart Contracts

### 6.3.1. Issue_cert Smart Contract

Upon receiving the CSR request *Issue_cert*, the smart contract internally verifies the provided Ethereum address and proceeds to verify the ECDSA signature using the *ecrecover* function from Solidity's standard library. Notably, *ecrecover* returns the Ethereum address associated with the signature instead of the public key. To authenticate the signature, the system reconstructs the Ethereum address from the given public key and compares it with the address from *ecrecover*. Subsequent to this comparison, the system conducts a verification of the domain name, country, state, and additional information. The validated CSR data are then stored within a key-value mapping storage structure, wherein the Ethereum address serves as the key and the CSR details constitute the associated value.

Subsequently, the controller activates the *Certificate* smart contract for creating the certificate related to this request and informing introducers for verification. Additionally, the *Issue_cert* smart contract is equipped with several read functions, which permit other smart contracts to access the CSR information via the Ethereum address. For instance, the *Certificate* smart contract utilizes these functions to generate X.509 certificates, whereas the *Verification* smart contract leverages them to provide comprehensive CSR information to members of the key-ring network.

Figure 8b illustrates a selection of the read and write functions available in the Remix IDE, as provided by the *Issue_cert*, and Figure 9 describes the transaction details.

### 6.3.2. Certificate Smart Contract

The *Certificate* smart contract pulls the required information from the *Issue_cert* through its read functions, making it accessible to the controller. Following this preparation, the controller is notified to generate the digital certificate conforming to the X.509v3 standard. The controller utilizes a standard Node.js library called node-merge, which efficiently generates the digital certificate by leveraging the data sourced from the Certificate smart contract. Moreover, the contract archives the certificate metadata on the Ethereum blockchain. These metadata include information such as the certificate ID, expiration date, requester's public key, certificate status, Ethereum addresses, and the signatures of introducers.

Upon certificate creation, it is ready for endorsement by the introducers. Consequently, the controller broadcasts the details of certificate creation to both the ring owner and the introducers to initiate the final phase of validation.

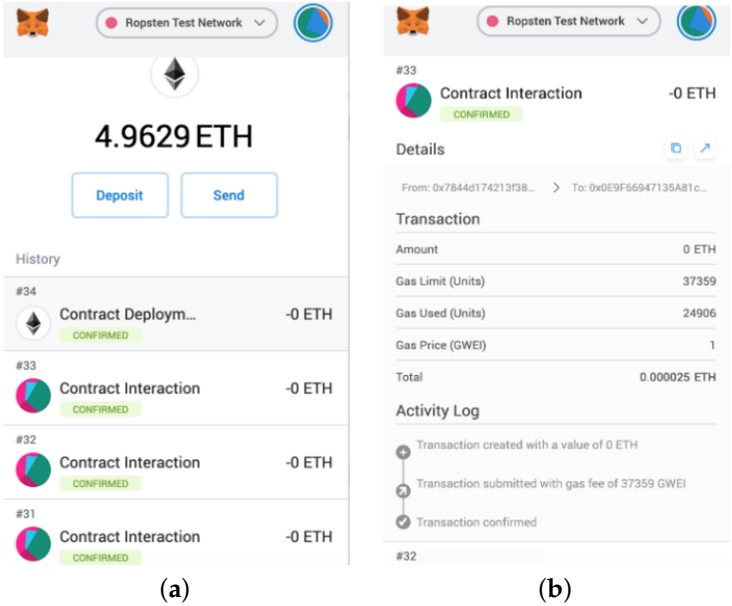

**Figure 9.** Transaction history (**a**) and single transaction details (**b**) from MetaMask.

### 6.3.3. Verification Smart Contract

The *Verification* smart contract serves to verify certificates and revocation information by the introducer of the key ring. Additionally, it plays a crucial role in constructing a key-ring network based on trust relationships. It features four main write functions—*addSigner*, *verifyAndSignCertInfo*, *verifyAndSignRevocationInfo*, and *setUserTrustLevel*, as shown in Figure 10. Any eligible introducer can be designated as the signer for this smart contract.

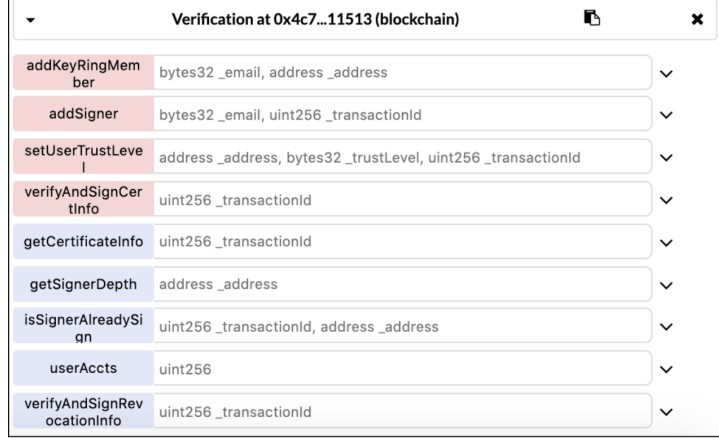

**Figure 10.** Read and write functions for verification using the Remix IDE.

Prior to endorsing any information, the introducer needs sufficient evidence to validate the transaction. The introducer utilizes the controller to facilitate communication with the requester through the designated email address. Upon successful manual verification, if the introducer is convinced of the legitimacy of issuing a certificate, they may proceed to utilize the certificate signing interface for certificate endorsement, as depicted in Figure 11a. At this point, the introducer is required to supply their private key for the signing of the certificate in addition to a transaction ID, which is necessary for interfacing with the *Certificate* smart contract. This interaction allows for the incorporation of the signature into the X.509 extension field. It also stores the address of the *Certificate* smart contract and specific verification details, including the introducer's identity, trust level, and final verification status. Upon receiving a verification outcome from the introducer, the smart

contract proceeds to assess a mathematical equation (Equation (1)) in order to determine the trust level for the requested CSR.

**User Information**

Domain Name: bCertDomain.com
Country : US
State : NM
Organization Name: New Mexico Tech
Organization Unit: Computer Science
Email Address: rhalder@mro.nmt.edu

**Note** Before Signing public key certificate for this information, please verify each information independently. If necessary contact with requester, or meet with him/her face to face, also can contact with administrative department of the company to verify identity.

**Sign Public Key Certificate**

Transaction Id

Enter your email

Signing Private Key(Will no be stored)

Choose File   No file chosen

Update Info

(**a**)

**User Information**

Domain Name: bCertDomain.com
Country : US
State : NM
Organization Name: New Mexico Tech
Organization Unit: Computer Science
Email Address: rhalder@mro.nmt.edu

**Add Trust Level**

Email

Trust Level

Reason to Trust

Transaction Id

Update Info

(**b**)

**Figure 11.** Public key certificate signing UI (**a**) and interface to add an introducer (**b**).

After multiple signatures, if the information is fully or marginally valid, the ring owner is responsible for finally signing the certificate and returning it to the requester using the controller system. Figure 12 shows an example of an SSL certificate from a browser.

Furthermore, introducers can use the interface, as depicted in Figure 11b, to introduce new introducers into the key-ring network. This interface initiates the execution of the *setUserTrustLevel* write function, which is responsible for calculating the trust level of the new introducer by employing the mathematical formulation presented in Equation (2). Upon successful calculation, if the trust level corresponds to either the fully or marginally trusted thresholds, as explained in Section 5.2.2, the smart contract then disseminates the information throughout the key-ring network.

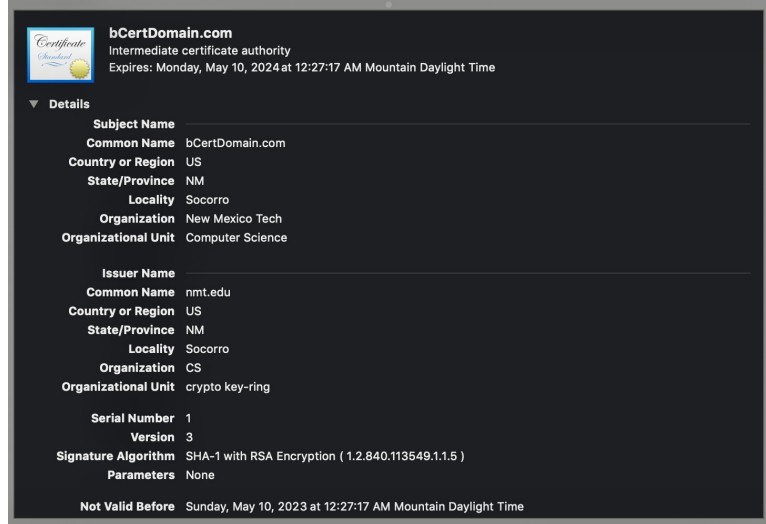

**Figure 12.** Generated SSL certificate.

*6.4. System Test*

We created a test environment to test the complete process of certificate creation and evaluate various operations related to Ethereum smart contracts. This environment was constructed using an array of software tools for Ethereum development, streamlining the

creation, deployment, and testing of smart contracts. MetaMask is a cryptocurrency wallet and browser extension that bridges web browsers and Ethereum, enabling us to interact with deployed applications, manage Ethereum accounts, and access balance and transaction details. Moreover, it can be configured to connect to a local blockchain or Ethereum test networks like Ropsten, Rinkeby, Kovan, and others.

Ganache is used to mimic a local Ethereum blockchain for testing, allowing smart contracts to be deployed swiftly without interacting with the actual Ethereum test network. It provides test accounts with private keys and addresses that can be imported into MetaMask for connection to the Ganache blockchain, streamlining the testing process. Additionally, alongside the Ganache local network, we used the Ropsten test network to create a testing environment that closely resembles the Ethereum mainnet.

For testing, we used a frontend UI and the Remix IDE, which helped execute the read and write functions of smart contracts. Figure 8a presents the CSR form interface and Figure 8b demonstrates the read and write functions of the *Issue_cert* smart contract. To issue a CSR, we utilized both the CSR form and the *issuingCSR* write function. The system internally executes additional read and write functions from *Issue_cert* to verify the requester's authenticity, as detailed in Section 5.2.2. It also enables existing introducers to conduct validations through the verification smart contract functions depicted in Figure 10 using the interface shown in Figure 11a. During the signing process, the *Verification* smart contract uses the *Certificate* smart contract functions to create an SSL certificate and insert signatures into the extension field. Figure 12 provides an example of a generated SSL certificate. Moreover, with the integration of the Remix IDE and the MetaMask browser extension, detailed transaction information becomes accessible. Figure 9a,b showcase how data visualization is rendered within MetaMask. This extension also facilitates the verification of detailed information through Etherscan.io.

## 7. System Evaluation and Limitations

In this section, our research framework is thoroughly evaluated in terms of security, scalability, interoperability, standardization, operational cost, and the underlying reasons. We list several important challenges that require better understanding to facilitate deployment efforts for decentralized PKI enhancements.

### 7.1. Security Evaluation

Our enhanced security framework addresses the challenges of existing centralized PKI systems, as detailed in Section 3. Furthermore, it offers better security solutions in certain areas compared to previously proposed blockchain-based PKIs. A detailed comparison is provided in Section 8. The key security enhancements our system introduces are as follows:

- Earlier research was primarily aimed at detecting attacks and fast certificate revocation, with less emphasis on detailed prevention strategies. Our research, however, prioritizes preventing malicious certification through an advanced verification system and a decentralized trust network. Public blockchains utilize cryptography to prevent unauthorized access, complemented by an initial authentication system designed for the same purpose. Furthermore, our verification system, upheld by members of a trusted key ring, ensures that certificates are not issued to malicious users.
- In our system, each certification and revocation action is meticulously recorded on the blockchain network, establishing a transparent and immutable public audit trail. This property provides a robust mechanism for tracking and identifying compromised certifications. As a result, the system is capable of executing faster revocations.
- The decentralized and tamper-resistant characteristics of blockchain significantly hinder attackers' ability to intercept and alter data exchanged between parties. Furthermore, numerous members are involved in creating and revocation of certificates, making MitM attacks nearly impossible.
- The inclusion of multiple signatures in the X.509v3 to issue a certificate significantly reduces the risk of CA compromise and the problem of a single point of failure.

Despite our significant advancements in security, vulnerabilities remain, particularly with smart contracts [43,44]. The immutable nature of smart contract code contrasts with the mutable state of its internal data, posing data security risks. Correcting smart contract errors is challenging and costly, necessitating careful adherence to coding standards. While we implement strict precautions, especially with Solidity, ongoing research into smart contract safety certification is essential for improving secure development and deployment. Additionally, the use of blockchain's transparent and immutable characteristics for certificate transparency raises user data confidentiality concerns, indicating a potential area for future enhancement.

### 7.2. Cost Analysis

Our system utilizes three specialized smart contracts, each designed for efficiency and minimal data storage to ensure cost-effectiveness. Key points include the following:

- **Simplicity and Efficiency:** By maintaining simple contract designs and employing Solidity's features, such as utilizing bytes over strings, preferring delegatecall where possible, and using bit operations, we aim to minimize gas costs.
- **Cost Calculation:**
  - **Low-Cost Operations:** Common tasks like request creation, signature additions, and revocations incur moderate costs.
  - **Free Operations:** Operations that do not modify blockchain data, such as CSR processing and data reading, are free.
  - **High-Cost Write Operations:** Functions that update blockchain data, especially storing CSR hashes and certificate metadata, are gas-intensive.

Due to the significant expenses of certain write operations, focused cost analysis and optimization can be performed to ensure the system's efficiency and cost-effectiveness.

### 7.3. Standardization and Interoperability

In the development of blockchain-based decentralized PKI systems, standardization and interoperability present significant challenges due to the lack of universally accepted standards. Key considerations for a new decentralized PKI system, as highlighted by Lesavre et al. [45], include the following:

- **Permission Type:** Choosing between a permissionless or permissioned blockchain.
- **Blockchain Type:** Evaluating the benefits of different blockchain implementations, like Ethereum or custom blockchains.
- **Trust Model Type:** Deciding between a hierarchical PKI or a WoT model.
- **Certificate Format:** Ensuring that formats like X.509 or custom formats support interoperability.
- **Revocation:** Incorporating certificate revocation as a critical security feature.
- **Implementable:** Demonstrating feasibility through proof-of-concept implementations.
- **Complexity and Cost:** Assessing and managing system complexity and operational costs.
- **Updateable Key:** Providing support for key updates for long-term sustainability.

In our proposed system, we implemented several key features to address these considerations:

- We adopted a permissionless blockchain for certificate transparency and suitability for large-scale applications.
- We chose Ethereum smart contracts for their decentralized infrastructure, facilitating the storage of certificate metadata to solve block traversal problems.
- We adopted a WoT trust model to eliminate centralized CAs and ensure a distributed verification system.
- We integrated multi-signature capabilities into the X.509v3 extension field, requiring minimal browser adjustments.
- We proposed a theoretical approach for expediting the revocation process.

- We provided a detailed system architecture with implementation specifics as a proof of concept. The system also supports the updateable key feature.

These steps ensure our system's interoperability and compliance with most standards, although our current setup does not address auto-scaling for larger networks, indicating areas for future development.

### 7.4. Scalability Evaluation

The rapid increase in participants within a system leads to blockchain network expansion, which underscores the scalability challenges in existing blockchain technologies. The blockchain trilemma [46] highlights a fundamental conflict among scalability, security, and decentralization within a public blockchain with numerous participants due to the trade-off between decentralization and scalability. As blockchain technology evolves, research is increasingly focused on enhancing scalability across its different layers.

However, ongoing development and research are addressing these scalability issues. Initiatives like ScalaCert [47] aim to improve scalability by integrating revocation data into certificates, eliminating the need for additional storage for CRLs. The Lightweight Scalable Blockchain (LSB) [48] employs a Distributed Time-based Consensus (DTC) algorithm for a faster mining process and features a distributed throughput system for adaptive scaling. Additionally, CBPKI [16] leverages cloud services like AWS for hosting PKI servers, addressing auto-scaling challenges.

Our study did not specifically address scalability issues in large networks, but certain adjustments can mitigate initial challenges. Ethereum 2.0 introduces a multi-phased enhancement plan. The first phase moves from the energy-heavy proof-of-work (PoW) model to a greener proof-of-stake (PoS) model and introduces sharding [49]. Through key infrastructure changes, the plan aims to significantly improve the network's scalability, energy consumption, and security. Additionally, scalability can be approached through horizontal scaling (adding more nodes), vertical scaling (upgrading individual node capabilities), and sharding (partitioning data for parallel processing). These strategies can benefit from existing cloud computing technologies for greater flexibility and efficiency.

While our system does not directly tackle scalability, it remains a critical area for future work, as we aim to develop a scalable, industry-standard decentralized PKI.

## 8. Discussion

Our proposal seeks to create a robust digital certification system with a decentralized PKI on the blockchain, addressing the single-point-of-failure issue and diminishing the risk of malicious attacks. Table 2 concisely outlines the methods employed to achieve our proposed objectives and the existing challenges we successfully tackled.

As we can observe from Table 2, the distributed trust model, inspired by the WoT approach, empowers authorized multiple introducers to validate and sign X.509v3 certificates. Involving multiple signatures to issue certificates enhances security, as breaching multiple ring members simultaneously becomes highly improbable. When establishing a distributed key ring, incorporating introducer depth limits the key-ring size and also prioritizes introducers who are in closer proximity to the ring owner.

The decentralized nature of the blockchain removes the need for a central authority, further reducing single-point-of-failure risks. Blockchain's inherent immutability safeguards data integrity, ensuring that records are tamper-proof and publicly auditable. This transparency, coupled with the involvement of multiple introducers, streamlines the revocation process and strengthens the system against attacks.

**Table 2.** Comprehensive feature set of the proposed system.

| Accomplishment | Approach | Existing Problem Addressed |
|---|---|---|
| Decentralized PKI | Create a blockchain-based decentralized PKI, utilizing a distributed trust model for enhanced security in the certification process | Centralized PKI limitations |
| Build distributed trust model | Employ a web of trust model to build a key ring with trusted members to work as CAs | Single point of failure |
| Introduce multiple introducers | Empower multiple authorized introducers to verify and sign X.509 certificates | Single point of failure, MitM attacks |
| Prioritized trusted introducers | Incorporating the concept of introducer depth helps limit the key ring's size and gives precedence to the introducer in closer proximity to the ring owner | |
| Minimal modifications on the browser side | Instead of changing the existing certificate format, we leverage the extension field in X.509v3 to incorporate additional signatures | |
| Information publicly visible | The nature of blockchain's distributed and append-only data entry helps to keep the certificate information in the public ledger | Lack of transparency |
| Faster revocation | Data transparency facilitates the detection of misbehaving CAs and the decentralized verification process and expedites the revocation of certificates | Convoluted revocation procedures |
| Data integrity | Blockchain's immutability ensures the integrity of stored data, making it tamper-proof | Data forgery attack |
| Refine certification algorithm | Provide an enhanced algorithm and procedure to support multiple signatures in X.509v3 | |
| System implementation architecture | Provide complete implementation architecture with workflow and necessary technologies | |

In our detailed analysis, we meticulously presented our goals and accomplishments in Table 2. By comparing the features and limitations of existing schemas in Section 2, we identified a common trend among many blockchain-based systems, as detailed in Table 3, utilizing the following metrics:

- **Trust Model:** Evaluates the trust relationship for deciding certificate legitimacy.
- **Certificate Format:** Specifies the proposed certificate format, like X.509 or PGP.
- **Auto-Scalable:** Indicates whether the system features auto-scalability for large networks.
- **Certificate Transparency:** Reflects the visibility of issued certificates.
- **Browser Side Unchanged:** Describes the minimal modifications required for browsers and servers to align with the proposed schema.
- **Decentralized Trust:** Involves distributed trust calculation across multiple servers.
- **Advanced Verification:** Utilizes strong authentication methods for precise identity verification.
- **Prohibits Malicious Certification:** Offers enhanced security and a robust verification system.
- **No Centralized CA:** Operates without a centralized certificate authority.
- **Monitors CA Activity:** Allows domains or external systems to oversee all CA operations.

It is evident that most blockchain-based systems proposed for decentralized PKI approaches can achieve the fundamentals; however, they have not yet successfully prevented malicious certification. Additionally, some still rely on a central CA with limited interoperability. Through our security analysis and achievements, our system successfully addresses existing PKI vulnerabilities, including preventing malicious certification, swiftly detecting misconduct, and expediting the revocation process. These advancements significantly boost security levels compared to other studies in this domain.

**Table 3.** Feature comparison of blockchain-based PKIs. Note: ✓ = supports the feature; × = does not support the feature; PY = partially supports the feature; N/A = not applicable.

| Schema | Trust Model | Certificate Format | Auto-Scalable | Certificate Transparency | Browser Side Unchanged | Decentralized Trust | Advanced Verification | Prohibits Malicious Certification | No Centralized CA | Monitors CA Activity |
|---|---|---|---|---|---|---|---|---|---|---|
| Yakubov et al. [29] | Hierarchical | Hybrid X.509 | × | × | × | PY | × | × | × | PY |
| CertLedger [13] | Hierarchical | X.509 | × | ✓ | × | PY | × | × | × | ✓ |
| Hwang et al. [15] | Hierarchical | X.509 | × | ✓ | × | PY | × | × | × | ✓ |
| CBPKI [16] | Hierarchical | X.509 | ✓ | ✓ | × | PY | × | × | PY | N/A |
| CertChain [17] | Hierarchical | Custom | × | ✓ | × | PY | × | × | × | ✓ |
| CertCoin [18] | WoT | Custom | × | ✓ | × | ✓ | × | PY | ✓ | ✓ |
| CeCoin [19] | WoT | Custom | × | ✓ | × | ✓ | × | PY | ✓ | ✓ |
| DBPKI [28] | WoT | Custom | × | ✓ | × | ✓ | × | PY | ✓ | ✓ |
| SCPKI [30] | WoT | Custom | × | ✓ | × | ✓ | × | PY | ✓ | × |
| Meta-PKI [33] | Hierarchical | Custom | × | ✓ | × | PY | ✓ | × | PY | ✓ |
| Proposed system | WoT | X.509v3 | × | ✓ | ✓ | ✓ | ✓ | ✓ | ✓ | ✓ |

## 9. Conclusions and Future Work

The inherent centralization of existing PKI systems introduces significant vulnerabilities, as a compromised CA can issue unauthorized certificates and access sensitive information. The proposed decentralized PKI model leverages blockchain and WoT technologies, where the authentication process is distributed across multiple entities. We detail how the integration of blockchain and the WoT can offer diverse security levels within the PKI framework. The proposed architecture addresses critical weaknesses of conventional CA-based PKIs, such as a single point of failure, MitM attacks, lack of transparency, and exposure to malicious actors.

While our proposed PKI system is comprehensive, certain aspects require further refinement. The inclusion of multiple participants in the verification process and the addition of multiple signatures on the X.509 certificates may introduce computational overhead during certificate verification. Envision a scenario where all the verifiers involved in the authentication process are leaf nodes of a key-ring network, with considerably high depth levels, and none of them are fully trusted. In such instances, a significant number of additional signatures is necessary for verification. This worst-case scenario highlights the need for future research to optimize the performance of multiple-signature verification procedures, particularly concerning large key-ring networks. Furthermore, to ensure seamless operation within large blockchain networks, future work should focus on developing solutions that address blockchain scalability challenges and optimizing energy usage.

**Author Contributions:** D.S. and R.H. conceived the presented idea and the experimental plan. R.H., D.D.R. and D.S. verified the analytical methods, and R.H. and D.D.R. carried out the experiments. R.H. and D.D.R. contributed to the final manuscript and all authors discussed the results. All authors have read and agreed to the published version of the manuscript.

**Funding:** This research received no external funding.

**Institutional Review Board Statement:** Not applicable.

**Informed Consent Statement:** Not applicable.

**Data Availability Statement:** No new data were created or analyzed in this study. Data sharing is not applicable to this article.

**Conflicts of Interest:** The authors declare no conflicts of interest.

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
