# Peer review of "A Blockchain-Based Decentralized Public Key Infrastructure Using the Web of Trust"

_jcp, doi:10.3390/jcp4020010_

Round 1

Reviewer 1 Report

The use of blockchain for decentralized certificate validation in the proposed decentralized Public Key Infrastructure (PKI) introduces noteworthy considerations. However, it is imperative for the authors to provide a more comprehensive discussion regarding the significance of this approach.

Clarification on Performance Implications: The manuscript briefly touches on potential performance challenges, but it lacks a detailed exploration of how these challenges may impact the practical implementation of the proposed system. Authors should delve into the specifics of expected transaction volumes and processing times, shedding light on the system's real-world viability.

Addressing Energy Consumption Concerns: The energy consumption aspect is briefly mentioned. To strengthen the manuscript, the authors should elaborate on the potential environmental impact and discuss any strategies employed to mitigate energy-related concerns in the proposed blockchain-based PKI.

Privacy Considerations: The issue of privacy is acknowledged, but a more thorough analysis is needed. Authors should explicitly outline the measures taken to ensure user data confidentiality and any potential trade-offs made in implementing a blockchain-based solution for certificate validation.

Smart Contract Security: The manuscript mentions smart contract vulnerabilities briefly. A more in-depth exploration of potential security risks associated with smart contracts and the specific safeguards implemented to address these risks is crucial for a comprehensive understanding.

Standardization and Interoperability: The manuscript briefly notes the evolving nature of standards for blockchain-based PKI. A more thorough discussion is warranted, including any efforts made to align the proposed system with emerging standards and enhance interoperability.

Elaboration on Technical Costs: The manuscript acknowledges technical costs but lacks detail. Authors should provide a breakdown of these costs, highlighting the financial and resource investments required for implementing and maintaining the proposed blockchain-based PKI.

In summary, the authors are encouraged to elucidate the practical significance of their work by providing a more detailed and nuanced discussion of the aforementioned points. This would enhance the clarity and impact of the manuscript, making it more accessible to a broader 

The use of blockchain for decentralized certificate validation in the proposed decentralized Public Key Infrastructure (PKI) introduces noteworthy considerations. However, it is imperative for the authors to provide a more comprehensive discussion regarding the significance of this approach.

Clarification on Performance Implications: The manuscript briefly touches on potential performance challenges, but it lacks a detailed exploration of how these challenges may impact the practical implementation of the proposed system. Authors should delve into the specifics of expected transaction volumes and processing times, shedding light on the system's real-world viability.

Addressing Energy Consumption Concerns: The energy consumption aspect is briefly mentioned. To strengthen the manuscript, the authors should elaborate on the potential environmental impact and discuss any strategies employed to mitigate energy-related concerns in the proposed blockchain-based PKI.

Privacy Considerations: The issue of privacy is acknowledged, but a more thorough analysis is needed. Authors should explicitly outline the measures taken to ensure user data confidentiality and any potential trade-offs made in implementing a blockchain-based solution for certificate validation.

Smart Contract Security: The manuscript mentions smart contract vulnerabilities briefly. A more in-depth exploration of potential security risks associated with smart contracts and the specific safeguards implemented to address these risks is crucial for a comprehensive understanding.

Standardization and Interoperability: The manuscript briefly notes the evolving nature of standards for blockchain-based PKI. A more thorough discussion is warranted, including any efforts made to align the proposed system with emerging standards and enhance interoperability.

Elaboration on Technical Costs: The manuscript acknowledges technical costs but lacks detail. Authors should provide a breakdown of these costs, highlighting the financial and resource investments required for implementing and maintaining the proposed blockchain-based PKI.

In summary, the authors are encouraged to elucidate the practical significance of their work by providing a more detailed and nuanced discussion of the aforementioned points. This would enhance the clarity and impact of the manuscript, making it more accessible to a broader 

Reviewer 2 Report

  1. The methodology section would benefit from additional details on the Ethereum smart contracts used, including their design and operational logic. Innovative integration of blockchain technology and WoT for a decentralized PKI, enhancing security and trust.
    • Comprehensive system architecture and implementation details are provided, including smart contracts on the Ethereum blockchain. Experimentally validated the proposed system, demonstrating practical applicability and effectiveness.
  2. A deeper analysis of the trust evaluation mechanism's efficacy in various scenarios would strengthen the paper. Consider including case studies or simulations. The discussion section could be expanded to address potential scalability issues and the implications for larger networks. Therefore, authors need to enrich the discussion on the verification, security, and reliability of blockchain systems. These references can provide additional depth to the paper's exploration of blockchain technology, particularly in the context of enhancing the security and reliability of decentralized systems through advanced verification techniques and observing dynamic behaviors in such applications, such as "BlockASP: A Framework for AOP-based Model Checking Blockchain System", and "Study and Investigation of PKI-Based Blockchain Infrastructure ". They will discuss various aspects of blockchain technology and PKI systems in blockchain systems verification and dynamic behavior observation. In addition,
  3. the review would benefit from a more structured comparative analysis of the mentioned blockchain-based PKI systems. This could include a table summarizing the features, benefits, and drawbacks of each system in comparison to the proposed system. Such a comparison would help readers understand the positioning of the proposed system within the existing landscape.
  4. "ScalaCert: Scalability-Oriented PKI with Redactable Consortium Blockchain Enabled "On-Cert" Certificate Revocation" and "OSM: Leveraging model checking for observing dynamic behaviors in aspect-oriented applications". They provide a scalability-oriented PKI called ScalaCert that utilizes redactable blockchain for certificate revocation. It does not mention a blockchain-based decentralized PKI using a web of trust. 
    • The paper acknowledges additional overhead compared to traditional PKI systems without providing a detailed analysis of the trade-offs.
    •  
    • There is limited discussion on the scalability of the proposed system and its performance implications in large-scale deployments.
    •  
    • The paper could benefit from a more detailed comparison with existing decentralized PKI solutions to highlight the unique contributions more clearly.
    • Some sections could be more concise, particularly the background and related work sections, to focus more on the novel contributions of the paper.
    • Some figures lack clear legends or explanations, particularly Figure 2 and 3. Please clarify.
  1. Table 1's data could be presented more effectively with a discussion of its implications for the proposed system's performance.
  2.  
  3. The conclusion could more directly address future research directions and potential challenges in adopting the proposed system.

Round 2

Reviewer 1 Report

After carefully considering the revisions made by the authors in response to the reviewers' comments, I am satisfied with the changes. The authors have adequately addressed the concerns raised during the initial review process.

After carefully considering the revisions made by the authors in response to the reviewers' comments, I am satisfied with the changes. The authors have adequately addressed the concerns raised during the initial review process.

Reviewer 2 Report

By using a decentralized structure, the research addresses centralized PKI system flaws such single points of failure and transparency issues. Blockchain and Web of Trust architecture for certificate issuance and revocation is a unique solution that might considerably enhance digital certificate dependability and protection.

The system evaluation evaluates the proposed solution's security, cost, and scalability. The study is commendable for assessing the proposed system's pros and cons and suggesting improvements.

The varied ecology of digital certificates and the requirement for widespread adoption of new technologies make interoperability and standardization important. The paper's acknowledgment of the challenges of establishing a decentralized Public Key system (PKI) in the present system and its proposed solutions are laudable.

Good, the mathematical formulations used for computing trust levels and the decision-making process for certificate validation and revocation are well-articulated.

While the system's reliance on Ethereum blockchain and smart contracts is justified, there is room for exploration of alternative blockchain platforms that might offer better scalability or lower operational costs